# Role of Preoperative Ultrasound Shear-Wave Elastography and Radiofrequency-Based Arterial Wall Tracking in Assessing the Vulnerability of Carotid Plaques: Preliminary Results

**DOI:** 10.3390/diagnostics13040805

**Published:** 2023-02-20

**Authors:** Daniela Mazzaccaro, Matteo Giannetta, Fabiana Fancoli, Giulia Matrone, Nicoletta Curcio, Michele Conti, Paolo Righini, Giovanni Nano

**Affiliations:** 1Operative Unit of Vascular Surgery, IRCCS Policlinico San Donato, Piazza Malan, 1, San Donato Milanese, 20097 Milan, Italy; 2Department of Electrical, Computer and Biomedical Engineering, University of Pavia, 27100 Pavia, Italy; 3Department of Civil Engineering and Architecture, University of Pavia, 27100 Pavia, Italy; 4Department of Biomedical Sciences for Health, University of Milan, 20133 Milan, Italy

**Keywords:** carotid endarterectomy, preoperative diagnosis, ultrasound, shear-wave elastography, carotid plaque vulnerability, RF-based arterial wall tracking

## Abstract

We aimed at evaluating the ability of point shear-wave elastography (pSWE) and of a radiofrequency (RF) echo-tracking-based method in preoperatively assessing the vulnerability of the carotid plaque in patients undergoing carotid endarterectomy (CEA) for significant asymptomatic stenosis. All patients who underwent CEA from 03/2021 to 03/2022 performed a preoperative pSWE and an RF echo-based wall evaluation of arterial stiffness using an Esaote MyLab ultrasound system (EsaoteTM, Genova, Italy) with dedicated software. The data derived from these evaluations (Young’s modulus (YM), augmentation index (AIx), pulse-wave velocity (PWV)) were correlated with the outcome of the analysis of the plaque removed during the surgery. Data were analyzed on 63 patients (33 vulnerable and 30 stable plaques). In stable plaques, YM was significantly higher than in vulnerable plaques (49.6 + 8.1 kPa vs. 24.6 + 4.3 kPa, *p* = 0.009). AIx also tended to be slightly higher in stable plaques, even if it was not statistically significant (10.4 + 0.9% vs. 7.7 + 0.9%, *p* = 0.16). The PWV was similar (12.2 + 0.9 m/s for stable plaques vs. 10.6 + 0.5 m/s for vulnerable plaques, *p* = 0.16). For YM, values >34 kPa had a sensitivity of 50% and a specificity of 73.3% in predicting plaque nonvulnerability (area under the curve = 0.66). Preoperative measurement of YM by means of pSWE could be a noninvasive and easily applicable tool for assessing the preoperative risk of plaque vulnerability in asymptomatic patients who are candidates for CEA.

## 1. Introduction

Despite the dramatic improvement in medical therapy and surgical techniques, in the industrialized world, ischemic stroke remains a major public health burden as the first cause of long-term disability. Severe internal carotid artery stenosis accounts for a major cause of all ischemic strokes [1]. Carotid endarterectomy (CEA) has been established as safe and effective for reducing this risk by randomized controlled trials that were conducted more than 20 years ago. Current guidelines for the treatment of carotid stenosis are still based on the evidence reported from those trials and recommend surgical revascularization in patients with a high degree of carotid stenosis. However, in asymptomatic patients, the benefit of CEA remains equivocal [2].

Clinical practice suggests that, at the same degree of stenosis, the characterization of the plaque is a major determinant of stroke risk, since vulnerable plaques can more easily cause cerebrovascular events [3].

Consequently, characterization of the plaque composition and its proneness to rupture is of crucial importance for preoperative risk assessment and treatment strategies.

Different biomarkers, such as circulating microRNA, and clinical features have been investigated as potential predictors of plaque composition, but none of them has emerged as a major determinant for plaque vulnerability [4].

Recently, ultrasound elastography has been increasingly used as a noninvasive appealing and promising tool that could help the characterization of plaque composition and its degree of stability [5].

Among the different ultrasound elastography techniques, strain and shear-wave elastography have been recently introduced for the assessment of the mechanical properties of the vascular system [6,7].

Strain elastography works with manual compression of the skin with the transducer on the region of interest. The tissue strain is then measured relative to the surrounding tissue and gives back a color-coded map that can overlay on the B-mode image. Strain elastography gives a qualitative assessment of the tissue strain [8].

On the other side, shear-wave elastography employs an acoustic radiation force impulse (ARFI) sequence to generate shear waves that propagate perpendicular to the ultrasound beam, causing transient displacements. The distribution of shear-wave velocities at each pixel is directly related to the measure of the tissue’s elastic properties.

ARFI elastography is further subdivided into point shear-wave elastography (pSWE) and multidimensional shear-wave elastography (2D- and 3D-SWE). While the 2D shear-wave elastography technique measures a large region of interest, in which a color-coded elastogram is obtained, pSWE measures only a fixed area (approximately 5 mm × 10 mm).

The application of ultrasound elastography techniques to the vascular system, and particularly for the assessment of elastic properties of the carotid tissue, is based on the rationale that higher stiffness regions correspond to the presence of calcifications, while lower stiffness values may indicate the presence of soft and/or hemorrhagic tissue, both features of vulnerable plaques [9].

The results obtained from strain elastography, however, may be more operator-dependent than those obtained from shear-wave elastography, which in turn may affect the correct evaluation of a carotid plaque [10]. Therefore, shear-wave elastography may provide a more precise quantification of the carotid stiffness; in particular, pSWE may be performed at a certain depth that fits better to the application to the carotid artery if compared to multidimensional shear-wave elastography.

Besides the data obtained from ultrasound elastography, the measurement of carotid stiffness during conventional duplex ultrasound may provide additional information for individual risk stratification [11]. These measurements can be performed noninvasively using radiofrequency (RF)-echo-based arterial wall tracking.

The literature, however, lacks data about the application of pSWE and of RF-echo-based arterial wall tracking in the assessment of carotid plaque stiffness and the evaluation of plaque vulnerability. Furthermore, data of histological validation of these techniques and methods are also still needed.

The aim of this study was then to evaluate the ability of pSWE, also supported by parameters derived from RF-echo-based arterial wall tracking in preoperatively assessing the presence of a vulnerable carotid plaque in patients undergoing CEA for asymptomatic stenosis.

## 2. Materials and Methods

This prospective monocentric study was approved by the Ethics Committee of San Raffaele Hospital on 20 June 2019 (110/int/2019) and registered on ClinicalTrial.gov (ClinicalTrials.gov Identifier: NCT05566080).

All patients signed an informed consent form for participation in this research.

From March 2021 to March 2022, all patients who had an asymptomatic carotid stenosis of 70–99% according to European Carotid Surgery Trial ECST [12] measurement at duplex ultrasound scan and who were candidates for CEA were consecutively enrolled.

Exclusion criteria were the presence of medical conditions limiting expected survival to less than 1 year, patients with significant uncontrolled or unstable medical conditions (heart failure or angina pectoris class NYHA III-IV, cardiac surgery in the previous 30 days, left ventricular ejection fraction <30%, severe chronic obstructive pulmonary disease, myocardial infarction in the previous 30 days, coronary heart disease with revascularization indication, that is, the common trunk or more than two coronary vessels), the presence of a tracheostomy, the presence of a paralysis of the laryngeal nerve contralateral to the carotid stenosis, women of childbearing potential, the inability to give informed consent, and patients with a medical history of stroke/TIA within the previous 6 months.

### 2.1. Ultrasound Elastography Evaluation

Before surgery, all patients underwent preoperative quantitative pSWE, using an Esaote MyLab ultrasound system (EsaoteTM, Genova, Italy), equipped with a 7.5 MHz linear array probe working at 7.5 MHz and using the Q-Elaxto software package. The same system and probe were also employed to measure parameters related to arterial stiffness through RF-echo-based wall tracking [13] using the Quality Arterial Stiffness (QAS) software package.

Systolic and diastolic brachial pressure were recorded noninvasively in all patients and entered into the software.

The QAS was performed with the patient in a supine position and with a slight neck extension, on the side of the carotid stenosis. The linear probe was placed along a longitudinal axis on the distal part of the common carotid artery just below the origin of the atherosclerotic plaque, at least 10 mm far from the origin of the bulb, strictly perpendicular to the ultrasound beam, with both walls clearly visualized (Figure 1). Then, the QAS algorithm was run, with automatic real-time measurement of the change in diameter of the vessel walls between the systolic and diastolic phases, caused by the traveling blood pressure wave originating by heart pumping (Figure 2). The local carotid pressure waveform is derived from brachial pressure values and vessel cross-sectional areas during the cardiac cycles [14]. All the measures are automatically calculated by the system starting from distension and pressure waveforms and provided in a report (Figure 3) [15].

In particular, the QAS report includes the following measures, which are derived from changes in diastolic or systolic vessel area/diameter in relation to the local pressure [16] (see Appendix A):-Distensibility coefficient (DC), i.e., the absolute change in vessel diameter during systole for a given pressure change;-Compliance coefficient (CC), i.e., the relative change in vessel diameter during systole for a given pressure change;-Alpha stiffness (α), which is the elastic coefficient of the vessel;-Beta stiffness (β), which is the elastic coefficient normalized on the diameter;-Pulse-wave Velocity (PWV);-Augmentation index (AIx).

Besides these measures, the QAS usually calculates other parameters such as augmented pressure (AP), the isovolumic contraction period (ICP), and the ejection duration (ED), which, however, were not included in our study.

The Q-Elaxto evaluation was performed with the patient in the same position as for QAS. The linear probe was placed along the longitudinal axis on the side of the carotid stenosis to evaluate quantitatively the stiffness of the carotid plaque, in terms of Young’s modulus (Figure 4).

The data derived from both these evaluations were correlated with the outcome of the intraoperative morphologic macroscopic and microscopic analysis of the carotid plaque removed during CEA.

According to the criteria described by Lovett et al. [17], vulnerable plaques were defined by the presence of at least 1 feature among large intraplaque hemorrhage, ulceration, large necrotic/lipidic core (≈>25% of total area), ruptured or thin (<65 μm) fibrous cap, inflammatory cell infiltration, and neovascularization of the plaque.

### 2.2. Statistical Analysis

Based on the results obtained by Garrard et al. [18], who found that unstable plaques had a mean Young’s modulus of 50.0 ± 19.6 kPa while stable plaques had mean values of 79.1 ± 33.8 kPa, assuming to have an alpha error of 0.05, we calculated that the enrollment of at least 10 patients per group would be enough to achieve a statistical power of 90%.

The obtained results were analyzed using the statistical software Stata^®^16.1 (StataCorp LLC, College Station, TX, USA). The normality of the distribution of continuous variables was tested by the Shapiro–Wilk test. Continuous variables are reported as mean ± 2 standard deviation (SD) in the case of Gaussian distribution; otherwise, the median and the interquartile range (IQR) are reported. Categorical variables are reported as numbers and percentages. The two-tailed Student’s *t*-test and the one-way ANOVA test were used to evaluate the differences in results between the group of patients with a vulnerable plaque and the ones with a stable plaque. Logistic regression analysis was also performed to evaluate if any preoperative factor could significantly affect the measurements obtained with the QAS and Q-Elaxto evaluations in the studied population. Values of *p* < 0.05 were considered statistically significant.

## 3. Results

A total of 63 patients were enrolled in the study within the analyzed period; 20 of them (31.7%) were female. The patients’ mean age was 74.5 ± 6.6 years.

As described in Table 1, patients were mainly affected by hypertension (90.5%) and dyslipidemia (88.8%). About half of them had a history of previous or current smoking (55.5%).

All patients were already taking statins and antiplatelet agents at the time of surgery. The plaque analysis revealed the presence of vulnerable plaques in 33 patients (52.4%). The demographic and clinical characteristics of both groups of patients, as reported in Table 1, showed a similar representation of cardiovascular risk factors between the two groups, except for a higher representation of the female sex in the group with stable plaque (50% vs. 15%, *p* = 0.01).

When comparing the data derived from the preoperative QAS evaluation between the two groups, patients who had a stable, calcified plaque showed values indicative of a general trend toward a greater stiffness than vulnerable, soft plaques, even if the *p* values were not statistically significant.

Particularly, both the distensibility coefficient and the coefficient of compliance tended to be lower in stable plaques (Table 2), while both the α coefficient and the β coefficient tended to be higher, even if all values did not reach statistical significance. AIx also tended to be slightly higher in patients with stable plaques than in those with vulnerable plaque, even if it was not statistically significant (10.4 + 0.9% vs. 7.7 + 0.9%, *p* = 0.16). The PWV was similar (12.2 + 0.9 m/s for stable plaques vs. 10.6 + 0.5 m/s for vulnerable plaques, *p* = 0.16).

Nevertheless, at pSWE evaluation, Young’s modulus was found to be significantly higher in patients with stable plaques (49.6 + 8.1 kPa vs. 24.6 + 4.3 kPa, *p* = 0.009). For the measurement of Young’s modulus, values >34 kPa had a sensitivity of 50% and a specificity of 73.3% in predicting plaque nonvulnerability (accuracy = 62.9%, area under the curve (AUC) = 0.66 ± 0.03, Figure 5).

At logistic regression analysis, none of the obtained measures was significantly affected by any of the preoperative factors in the overall cohort of patients.

## 4. Discussion

The application of ultrasound techniques for the functional assessment of mechanical properties of vessel walls in the understanding of major atherosclerotic diseases has progressively gained increasing appeal among medical researchers [19] thanks to its wide in vivo applicability [20].

Particularly, when applied to the carotid district, the preoperative evaluation of a carotid plaque could be enriched by further precious information that goes beyond the simple assessment of the degree of stenosis with the B mode, especially for neurologically asymptomatic patients in whom the benefit of CEA is questionable in the case of ongoing best medical treatment [21].

Nevertheless, current clinical practice suggests that not all “high-degree” carotid stenosis have the same risk of cerebrovascular events, and this risk particularly is linked to the morphology and to the composition of the carotid plaque, which define the plaque itself as “vulnerable” or “stable” [22]. Since “vulnerable” plaques have been associated with an increased risk of stroke [23], the preoperative identification of vulnerable lesions is crucial for a proper stratification of the operative risk, with a view to providing a stronger justification for the surgical treatment. Currently, the identification of plaque vulnerability features can only be performed with certainty through histological examination after the surgical removal of the plaque [4]. It therefore appears necessary to find a method that can be used in the preoperative phase and that provides information on the characterization of the plaque as near as possible as what can be performed with histological postoperative evaluation.

Particularly, the grayscale median B-mode technique has been studied in the literature as a potential tool that can provide further information about the presence of either hard hyperechoic or soft hypoechoic regions within the plaques for a deeper evaluation of carotid plaque composition. Lower grayscale median values in fact may correlate with the presence of a vulnerable plaque [24], but conflicting data exist about its real clinical utility [25].

Another potentially exploitable ultrasound technique is contrast-enhanced ultrasound (CEUS), which can detect neovessels and microvascularization within the plaque, which are signs of plaque inflammation and intraplaque hemorrhage and therefore of plaque vulnerability [26]. Nevertheless, the use of CEUS in routine clinical practice is limited by the need for intravenous contrast agents, with an associated risk of allergic reactions and increased costs.

Indeed, the potential application of elastography ultrasound techniques in the carotid district, and in particular of SWE, is now becoming more and more a focus of interest. Recently, Pruijssen et al. provided a literature overview of the last ten years on the diagnostic value of SWE in atherosclerotic diseases and identified 19 studies demonstrating the ability of SWE to assess plaque vulnerability, mainly based on symptomatology and echogenicity [7].

Sivasankar et al. performed a prospective, observational, comparative study on 60 patients with atherosclerotic plaques, who were divided into two groups of 30 each, based on history of stroke [27]. They found significantly higher stiffness values in patients without history of stroke, concluding that SWE can be used as a tool for the early detection of vulnerable carotid artery plaques.

Zhang et al. in their study assessed the stiffness of carotid plaques, comparing the quantitative measurements obtained with shear-wave elastography to those obtained with grayscale imaging [28]. They found that hyperechoic carotid plaques showed higher stiffness values as compared to hypoechoic plaques, concluding that shear-wave elastography imaging was a noninvasive, reproducible, and reliable method for the assessment of carotid plaque.

Nevertheless, data about histologic validation of SWE have been scarcely reported up to now. Pruijssen et al. in their review identified only two studies that compared the results of SWE with those obtained from the histologic analysis of human carotid plaques [18,29]. More in detail, Di Leo et al. [29] showed that Young’s modulus values had a sensitivity of 87.1% and a specificity of 66.7% in detecting histologically vulnerable plaques of 43 consecutive patients, with an AUC of 76.9%. Similarly, Garrard et al. [18] found that in a cohort of 25 patients, Young’s modulus values of unstable plaques were significantly lower than those of stable plaques (50.0 kPa vs. 79.1 kPa; *p* = 0.027), particularly when intraplaque hemorrhage, thrombus, or increasing numbers of foam cells were recognized at histology.

Furthermore, there are scarce data in the literature about the specific use of pSWE for the evaluation of the carotid plaques. Point SWE has been used for the evaluation of liver fibrosis [30] and of thyroid nodules, and the data reported in the literature to date confirm that pSWE has similar sensitivity and specificity compared to 2D-SWE when measuring tissue stiffness [31].

Nevertheless, when applied to the carotid artery, pSWE may have the advantage of measuring a fixed, smaller area if compared to 2D-SWE, which in turn may give a more precise evaluation of the plaque stiffness without the potential influence of other confounding factors from the nearby tissues.

Our results on 63 patients provide a histologic validation that further supports the potential role of pSWE in the preoperative risk stratification of patients with carotid plaques.

Besides pSWE, RF-based arterial wall tracking can provide additional information about arterial stiffness, which is a well-recognized marker of increased cardiovascular mortality and morbidity [32].

Arterial stiffness, measured with PWV, has been proposed as a prognostic factor of silent cerebral ischemic lesions in asymptomatic patients with atherosclerotic carotid plaques [33]. Furthermore, Li et al. in their pilot study about 11 patients diagnosed with moderate (>50%) to severe (>80%) carotid artery stenosis investigated the potential of PWV coupled with imaging in characterizing the composition of carotid plaques [34].

These findings suggest that the information on carotid wall stiffness parameters derived from the radiofrequency echo-tracking-based method can provide support to ultrasound elastography for the evaluation of plaque composition.

As a novelty, our study showed that the parameters derived from the arterial wall tracking method may add further data for a proper assessment of asymptomatic patients undergoing CEA, even if the values did not differ in a statistically significant way between patients with vulnerability and those with stable plaques. This was probably due to the small sample size, which is the main limitation of our study.

Notably, distensibility and compliance tended to be lower in patients who had a stable, calcified plaque if compared to patients with soft, vulnerable plaques. Additionally, the augmentation index, which is strongly correlated with arterial distensibility and can be considered a good surrogate for the evaluation of arterial stiffness [35], tended to be higher in patients with stable plaques.

Further studies with a larger sample size may add support to the usefulness of pSWE and RF-based tracking wall techniques in the preoperative assessment of asymptomatic patients who are candidates for carotid endarterectomy for significant stenosis.

## 5. Conclusions

The preoperative value of Young’s modulus, assessed by means of pSWE, can reliably predict noninvasively the preoperative risk of plaque vulnerability in asymptomatic patients who are candidates for CEA. In particular, in our study on 63 patients, values >34 kPa had a good specificity in predicting plaque nonvulnerability.

## Figures and Tables

**Figure 1 diagnostics-13-00805-f001:**
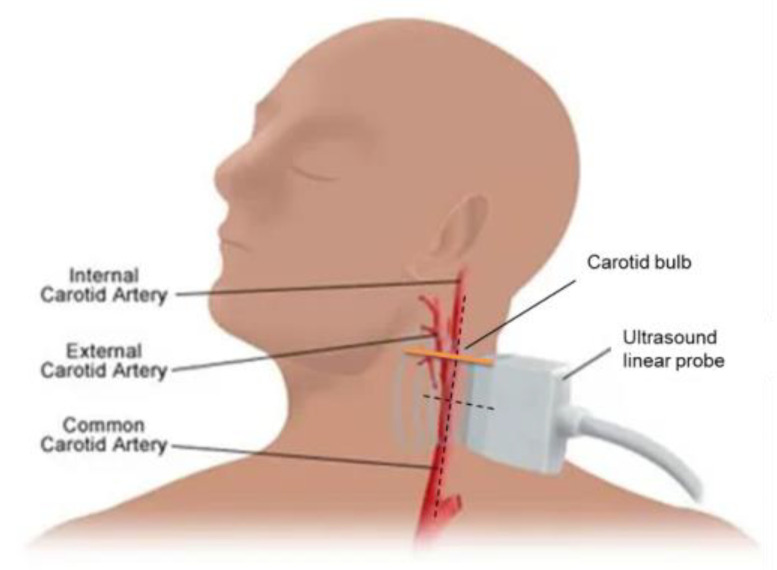
Position of the patient for the ultrasound evaluation (both for QAS and for Q-Elaxto). Note the position of the linear array on the vessel’s longitudinal axis (black dashed line), just below the carotid bulb (orange line), and the ultrasound beam is perpendicular to that.

**Figure 2 diagnostics-13-00805-f002:**
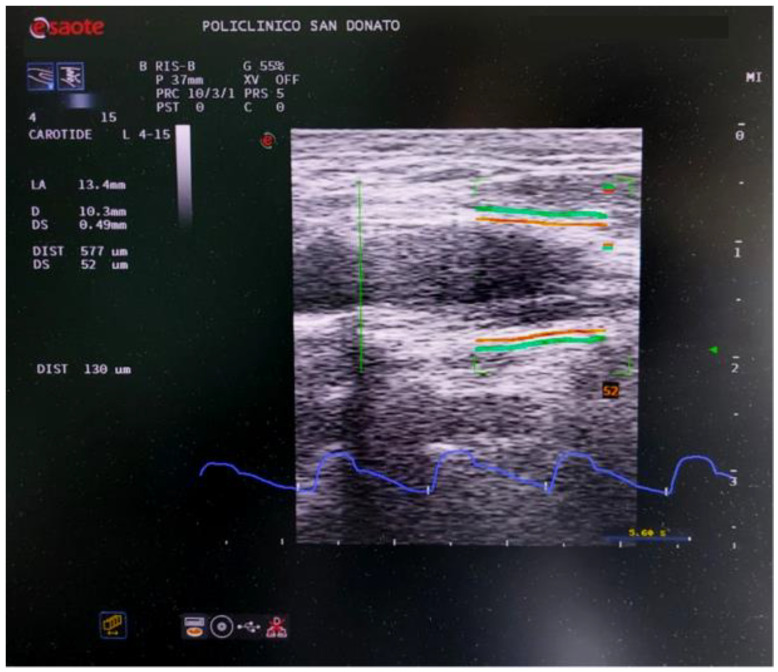
Screenshot of the QAS evaluation of the carotid artery on the diseased side. The red lines represent the vessel wall average diameter tracking. The green lines are associated with the wall distension. The real distensibility represented by the green line movement is “amplified” giving a fast estimation of the vessel’s elastic properties. The velocity curve over time is shown in blue below the ultrasound image.

**Figure 3 diagnostics-13-00805-f003:**
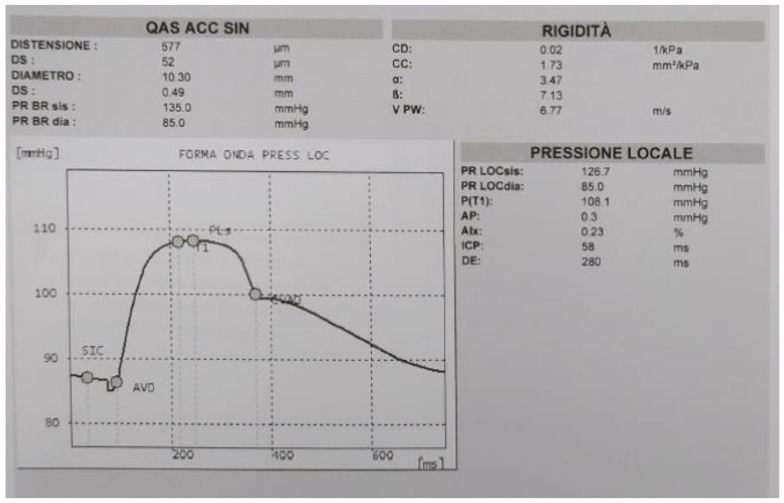
Report of QAS with reported measurements and the plotted graph of local pressure waveform versus time. The mean distension with standard deviation and the mean diameter with standard deviation are measured (up left side), along with some parameters of stiffness (on the right side) that are explained in the text and in Appendix A.

**Figure 4 diagnostics-13-00805-f004:**
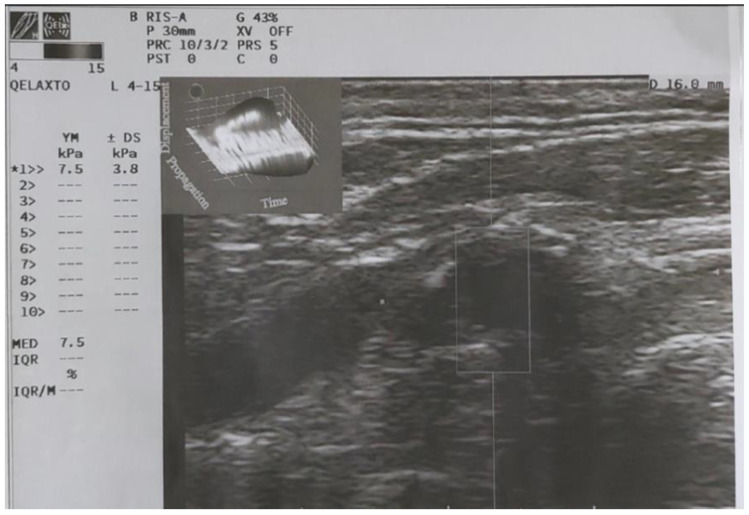
Q-Elaxto evaluation of the carotid plaque with measurement of Young’s Modulus.

**Figure 5 diagnostics-13-00805-f005:**
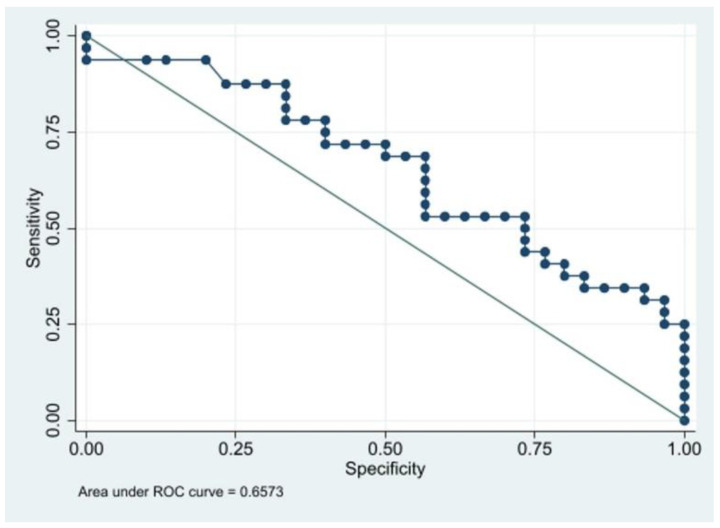
ROC curve of Young’s modulus for predicting plaque nonvulnerability.

**Table 1 diagnostics-13-00805-t001:** Details of preoperative characteristics of the analyzed cohort of patients. Values are represented as number (%) or mean + 2 standard deviations. Significant *p* values are reported in bold in the third column.

Total *n* = 63	Vulnerable (*n* = 33)	Stable (*n* = 30)	*p* Value
**Female sex**	5 p (15.1%)	15 p (50%)	**0.01**
**Age**	75.4 + 7.2	73.6 + 6.9	0.81
**BMI**	25.3 + 3.8	25.8 + 4.3	0.53
**Smoking habits**	Active: 5 p (15.1%)	Active: 8 p (26.7%)	0.49
	Past: 16 p (48.5%)	Past: 19 p (63.3%)	
**Hypertension**	27 p (81.8%)	30 p (100%)	0.56
**Dyslipidemia**	26 p (78.8%)	30 p (100%)	0.87
**CAD**	6 p (18.2%)	10 p (33.3%)	0.35
**COPD**	1 p (3%)	3 p (10%)	0.35
**Diabetes Mellitus**	8 p (24.2%)	11 p (36.7%)	0.56

p = patients; BMI: Body Mass Index; CAD: Coronary Artery Disease; COPD: chronic obstructive pulmonary disease; ASA: acetyl-salicylic acid.

**Table 2 diagnostics-13-00805-t002:** Comparison of the data derived from the preoperative ultrasound elastography between the group of patients with vulnerable plaque and those with stable carotid plaque. Significant *p* values are reported in bold in the third column.

Parameter	Vulnerable	Stable	*p* Value
**Young’s Modulus**	24.6 + 4.3 kPa	49.6 + 8.1 kPa	**0.009**
**Distensibility Coefficient**	329.1 + 142.9 µm	297.4 + 176.1 µm	0.44
**Coefficient of Compliance**	0.66 + 0.35 mm^2^/kPa	0.52 + 0.3 mm^2^/kPa	0.09
**α-Index**	8.7 + 4.4	12.3 + 12.7	0.14
**β-Index**	17.7 + 8.8	24.9 + 25.5	0.14
**Pulse-Wave Velocity**	10.6 + 0.5 m/s	12.2 + 0.9 m/s	0.16
**Augmentation Index**	7.7 + 0.9%	10.4 + 1.7%	0.16

## Data Availability

Not applicable.

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
