# Peer review of "Role of Preoperative Ultrasound Shear-Wave Elastography and Radiofrequency-Based Arterial Wall Tracking in Assessing the Vulnerability of Carotid Plaques: Preliminary Results"

_diagnostics, 2023, doi:10.3390/diagnostics13040805_

Round 1

Reviewer 1 Report (Previous Reviewer 2)

The manuscript aimed to evaluating the ability of point shear-wave elastography (pSWE) and radiofrequency (RF) echo tracking based method, in assessing preoperatively the vulnerability of the carotid plaque in patients undergoing carotid endoarterectomy (CEA) for significant asymptomatic stenosis. Finally concluded that preoperative measurement of YM by means of pSWE could be a non-invasive and easily applicable tool for assessing the preoperative risk of plaque vulnerability in asymptomatic patients who are candidates for CEA.

There are still several comments need to be assessed.

1.     Overall, the cases in this study is quite not enough. If necessary, the process for calculating the necessary cases for the study need to be listed.

2.     The unit for each factor in Table 1 need to be added.

3.     The ROC curve of Young's modulus for predicting plaque non-vulnerability need to be shown.

Author Response

The manuscript aimed to evaluating the ability of point shear-wave elastography (pSWE) and radiofrequency (RF) echo tracking based method, in assessing preoperatively the vulnerability of the carotid plaque in patients undergoing carotid endoarterectomy (CEA) for significant asymptomatic stenosis. Finally concluded that preoperative measurement of YM by means of pSWE could be a non-invasive and easily applicable tool for assessing the preoperative risk of plaque vulnerability in asymptomatic patients who are candidates for CEA.

There are still several comments need to be assessed.

  1. Overall, the cases in this study is quite not enough. If necessary, the process for calculating the necessary cases for the study need to be listed.

We thank the Reviewer for the valuable comment. Based on the results obtained by Garrard et al. [18], who found that unstable plaques had a mean Young’s Modulus of 50.0+19.6 kPa, while stable plaques had mean values of 79.1+33.8 kPa, assuming to have an alpha error of 0.05, we calculated that the enrollment of at least 10 patients per group would be enough to achieve a statistical power of 90%. This information was added as suggested by the Reviewer (please see P6 L183-186).

  1. The unit for each factor in Table 1 need to be added.

We thank the Reviewer for the valuable comment. The unit for each factor in Table 1 has been added as suggested (please see P7).

  1. The ROC curve of Young's modulus for predicting plaque non-vulnerability need to be shown.

Figure 5 showing the ROC curve of Young's modulus for predicting plaque non-vulnerability has been added, as suggested by the Reviewer (please see P8).

Reviewer 2 Report (Previous Reviewer 1)

Thank you for the modifications and additions.

Author Response

We are grateful to the Reviewer for the time and effort spent for the revision of the paper.

Round 2

Reviewer 1 Report (Previous Reviewer 2)

All the comments are well answered. And the revised version of manuscript is acceptable.

This manuscript is a resubmission of an earlier submission. The following is a list of the peer review reports and author responses from that submission.

Round 1

Reviewer 1 Report

The paper, in its current shape, presents major flaws:

(1) Overall, the distribution of content along the manuscript is not appropriate. The introduction is missing key information to prepare the reader with the basics to understand the rest of the manuscript. (2) The technical concepts of SWE and pSWE are used instinctively along the document. (3) The contribution and main novelty to the literature is missing. In the discussion it can be read that there the findings of this manuscript agrees with what is in the literature, however, the authors do not indicate what is the advance in the knowledge.

Specific flaws:

0. Abstract. CEA has not been introduced while other terms have.

1. Introduction: the motivation of using the two imaging techniques proposed, SWE and RF-echo based arterial wall tracking, is not fully elaborated. The reader could easily get lost as the basics of the techniques, the advantages against other methods and the working hypothesis for their application in vascular imaging are not well explained. Many of this information is incorrectly included in the discussion section.

2. Materials and methods. Overall, some key information to understand how the techniques are applied in the study are missing or in the discussion section.

2.1. There is repeated sentences/information at the beginning of the subsection.

Line 101: "different parameters can be computed", what parameters? mention then, or indicate that they will be shown later. Be specific. Add reference.

Explain both images. Little information is available in the caption of both Figures 1 and 2. The quality of both is poor.

A figure supporting the configuration of the experiment as indicated in the text (position of probe respecting the artery, are to be imaged, etc...) would improve the section.

The physical concept and meaning of the Alpha, beta, PWV and AIx parameters are not explained.

3. Results:

BMI, CAD and COPD are relevant to the study? if so, comment on their values.

Line 174-175: rephrase it to indicate that the young's modulus is affected by the gender, and quantify the effect. In the discussion section, an explanation to this effect should be provided.

4. Discussion

This section is not well structured. The order will confuse the reader. The section contains info about the motivation and techniques that should be in the introduction and methods sections, respectively. SWE and pSWE are used indistintively, however, in the literature they appear to be different. See Bamber et al. 2013: EFSUMB Guidelines and Recommendations on the Clinical Use of Ultrasound Elastography. Part 1: Basic Principles and Technology.

Line 188: ref 15 is about SWE, not pSWE.

Line 209: a reference is needed.

Paragraphs from 217 to 224 need to be grouped. 

In the middle of the discussion there is a massive block of generic information about elastography. From line 225 to 254. I would recommend to be more specific for the case of vascular elastography.

Line 259: ref 7 is also about SWE, mostly SuperSonic Imaging by Aixplorer.

What is the main contribution of the work to the literature? It is not clear what is the advance in knowledge respecting the current state-of-the-art. The study claims that it agrees with what has done before, however, the advance in knowledge is not stated. Additionally, the authors indicate that the number of patients might be not sufficient to provide strong statistically conclusions in the utility of the parameters that are not the Young Modulus.

Reviewer 2 Report

This manuscript submitted by  Mazzaccaro et al. investigated the ability of point shear-wave elastography (pSWE) and radiofrequency (RF) echo tracking based method in assessing the vulnerability of the carotid plaque in patients undergoing carotid endarterectomy preoperatively. This is an interesting contribution toward the monitoring of  carotid plaque. However, the manuscript needs some modifications.

1. Abstract - The full forms of CEA should be provided at their first appearance in the abstract.

2. Materials and Methods - The first paragraph in “2.1. Ultrasound-elastography evaluation” is similar to the second and third paragraph in meaning.

3. Results – What is the 95% confidence interval of the AUC of  Young's modulus for predicting plagque non-vulnerability?

4. Results – how was the result that “sex was the only preoperative factors that affected significantly the value of Young’s modulus among all the cohort of patients, with lower values in female patients (P=0.03).” obtained?

5. Results – Please add logistic regression analysis based on sex and ultrasound elastography parameters for predicting plagque non-vulnerability.